# An Innovative Approach to Prepare Liquid-Solid Dual-Phase Flowable Tritium Breeder with Low MHD Effect

**DOI:** 10.3390/ma16010406

**Published:** 2023-01-01

**Authors:** Kun Xu, Yanfei Qi, Bo Wang

**Affiliations:** 1Institute of Advanced Energy Materials and Devices, Beijing University of Technology, Beijing 100124, China; 2Institute of Metallurgy and Energy, North China University of Science and Technology, Tangshan 063210, China

**Keywords:** MHD effect, liquid tritium breeder, Li_2_TiO_3_, viscosity, liquid lithium

## Abstract

In present paper, a novel flowable tritium breeder is prepared by mixing the Li_2_TiO_3_ micro-powders and liquid GaInSn alloy, where GaInSn alloy is used to simulate the fluid behaviors of lithium-based liquid tritium breeder, forming a type of composite characterized by liquid-solid dual phase. In detail, the effects of the volume fraction of ceramic micro-powders on viscosity and conductivity of the composite in magnetic field are the focus. The XRD results prove that the obtained Li_2_TiO_3_ micro-powders contained Li_2_TiO_3_ phase without impurities. The results shows that once the magnetic field intensity exceeds the critical value, the viscosity of liquid GaInSn metal becomes significantly greater than that of liquid-solid dual-phase composites. Furthermore, the addition of Li_2_TiO_3_ micro-powders could effectively reduce the magneto hydro dynamic (MHD) fluid effect, and the dual-phase composites exhibit comparatively lower flow resistance under the strong magnetic field. Moreover, the conductivity of the tritium breeder composites decreases rapidly with the addition of Li_2_TiO_3_ micro-powders. The MHD pressure-drop-increasing rate decreases with the increase of viscosity, which indicates that the addition of Li_2_TiO_3_ micro-powders effectively reduces the MHD effect. The conductivity of the composites increased slightly and then remained stable after static placing for several tens of minutes. The present investigation provides a novel insight into the fabrication strategy of tritium breeder materials with low MHD effect.

## 1. Introduction

With the rapid development of social economy, the consumption of non-renewable energy such as coal and oil has long led to increasingly serious environmental problems. Nuclear fusion energy has become the preferred energy in the future because of its safety, cleanliness, rich fuel resources, and other advantages [1]. In the fusion reactor, tremendous energy is produced by nuclear fusion of deuterium (D) and tritium (T). Deuterium is abundant in nature, whilst tritium is rare on earth and is usually prepared by neutron irradiation with lithium-based alloys as raw materials [2,3,4]. In order to compensate for the tritium consumed in D-T fusion reaction, lithium-containing materials have been used to react with the neutron, which is the product during the reaction, thus realizing the so-called “tritium self-sustaining” process. In this process, a tritium breeder blanket is usually designed in the fusion reactor or fusion-fission hybrid reactor, where lithium-containing materials are important tritium-producing materials, namely tritium breeder materials. Tritium breeder materials can be generally divided into liquid tritium breeder materials and solid tritium breeder materials. Liquid tritium breeder materials are mainly liquid lithium and its alloys, such as Li_17_Pb_83_, Li_2_BeF_2_, Li_25_Sn_75_, etc. [5,6]. Liquid breeder materials were first employed in breeder blankets due to their advantages of easy tritium extraction, resistance to irradiation damage, high thermal conductivity, and easily achieved manufacturing and specification requirements [7]. However, three major problems hinder the wider application of liquid breeder materials: (1) magnetohydrodynamic (MHD) effects increase fluid resistance and reduce liquid Li mobility [8]; (2) the corrosion candle of liquid Li on cladding structural materials [9]; and (3) the inherent volatilization of liquid Li at operating temperatures leads to Li loss [10]. Numerical simulation is an important method to study the MHD effect of liquid metal cladding. The cladding pipeline structure in the practical application of nuclear fusion is very complex, but the research mainly focuses on the straight pipeline, such as round pipe and square pipe. Wang et al. [11] studied the induced current and velocity distribution in lithium lead square tubes and found that the Lorentz force is very large, and the side velocity was about four times that of the center velocity. Buhler et al. [12] studied the changes of flow velocity and Hardman number in the closed loop and found that with the increase of Hardman number, the flow velocity decreased rapidly. Solid tritium breeder materials include lithium silicate (Li_4_SiO_4_), lithium titanate (Li_2_TiO_3_), lithium titanate (Li_2_ZrO_3_), and lithium oxide (Li_2_O) [13,14,15]. Among those alloys, Li_2_TiO_3_ is recognized as one of the most promising solid tritium breeder materials and has attracted increasing scientific interests due to the considerable lithium atom density, low activation, excellent chemical stability, and good compatibility with structural materials [16] combined with acceptable tritium release performance at low temperature [14].

Conventional monophase tritium breeder materials can hardly satisfy the utilization requirement of the fusion reactor. First, although the liquid tritium breeder materials have the advantages of a high tritium breeding ratio and high lithium content [17], the MHD effect can cause intense flow resistance and thus seriously affects the flow behaviors of liquid tritium breeder materials. Further, the liquid tritium breeder materials easily corrode the cladding structural materials. On the other hand, the solid tritium breeder materials exhibit high chemical stability, safety, and non-corrosiveness without MHD effect [18,19]. However, the prominent problems hindering the development of these solid breeder materials include low lithium density, poor tritium release and heat transfer performance, easy blockage of tritium transport gas channel, and complex cladding structure.

In order to overcome the disadvantages of monophase liquid or solid tritium breeder materials, a novel flowable dual-phase tritium breeder material is prepared in present study for the first time, to the best of authors’ knowledge. The proposed flowable tritium breeder material fabricated by mixing liquid metal and solid ceramic micro-powders can not only suppress the MHD effect of the existing liquid metal or molten salt tritium breeder agent but also eliminate the problems of low tritium release efficiency, low heat transfer, fragility, and carrier gas channel blockage caused by lithium volatilization of the solid tritium breeder materials. In this paper, liquid GaInSn alloy is used to simulate the fluid behavior of lithium-based liquid tritium breeder, which is mixed with various volume fraction of Li_2_TiO_3_ ceramic micro-powders to form a liquid-solid dual-phase tritium breeder composite. The aim of the present study is to elucidate the effects of volume fraction of ceramic micro-powders on viscosity in magnetic field and on the conductivity of tritium breeder composites. The present study mostly concentrates on numerical simulation of liquid tritium breeders to study its MHD effect. In this paper, the relationship between viscosity and magnetic flux density of composite fluid under magnetic field is innovatively studied by experiments, and the linear relationship between MHD pressure drop and kinematic viscosity is explored. The result may provide direction to completely solve the problem of liquid and solid tritium breeder materials, which is beneficial to the future development of tritium breeder materials.

## 2. Experimental

### 2.1. Preparation of Li_2_TiO_3_ Micro-Powders

A novel sol-gel-hydrothermal method, which was developed on the basis of conventional sol-gel method and hydrothermal method, was used to prepare the Li_2_TiO_3_ micro-powders. The main procedures are as follows: (1) heating the 100 mL ethylene glycol at 140 °C oil bath and mixing 6.72 g LiOH.H_2_O with the above solution and stirring for a period of time, followed by cooling in air; (2) mixing the C_16_H_36_O_4_Ti (tetrabutyl titanate) with the above cooled solution by maintaining the molar ratio of 1:2.25; and (3) stirring the mixed solution until white gel precipitated. The hydrothermal reaction was performed in the hydrothermal reactor at 160 °C for 10 h, followed by drying in oven at 60 °C. Finally, the Li_2_TiO_3_ ceramic micro-powders were obtained by calcining at 600 °C for 1 h.

### 2.2. Preparation of Flowable of Liquid-Solid Dual-Phase Tritium Breeder Composite

Different certain volume fractions of Li_2_TiO_3_ micro-powders and liquid GaInSn alloy were mixed, crushed, and stirred repeatedly to obtain a uniform liquid-solid dual-phase tritium breeder composite.

### 2.3. Construction of Magnetic Field Viscosity Device

Referring to McTague et al. [20], a testing device for determining the variation of the viscosity in the applied magnetic field with different magnetic flux density was set up as shown in Figure 1. In Figure 1, the capillary tube is put in an external magnetic field, which is generated by the excitation coil. A maximum magnetic flux density of 2305 GS can be achieved. By measuring the time required for a certain volume of fluid to flow through the capillary under its own gravity, the kinematic viscosity (*η*), which is used to evaluate the fluidity of flowable tritium breeder, is calculated from Equation (1) [21]:(1)$$\eta =\frac{\pi r^{4}gh}{8VL}t,$$
where r is the inner diameter of the viscometer, g is gravitational acceleration, h is the height difference between the two tubes of the viscometer, V is the volume of the fluid, and L is the length of the capillary. $\frac{\pi r^{4}gh}{8VL}$ could be regarded as a constant; r is measured as 0.4 mm. t represents the time for a certain volume fraction of fluid flowing through the capillary under its own gravity.

### 2.4. Testing and Characterization

The morphology of Li_2_TiO_3_ micro-powders was observed by Hitachi scanning electron microscope (SEM, SU-70). The phase constitution of Li_2_TiO_3_ powders was determined by the X-ray diffractometer (XRD, Bruker D8 advance) with Cu radiation. The conductivity of the tritium breeder composites with various volume fractions of ceramic micro-powders was measured by conductivity meter.

## 3. Results and Discussion

### 3.1. Micromorphology and Structural Characterization of Li_2_TiO_3_ Micro-Powders

Figure 2a shows the SEM image of Li_2_TiO_3_ micro-powders. It was found that Li_2_TiO_3_ grains tend to be complete, the particle morphology is spherical, the sphericity is high, the boundary is very clear, and the agglomeration is light. Figure 2b shows the particle size distribution of Li_2_TiO_3_ micro-powders prepared by sol-gel-hydrothermal method. The spherical nanoparticles were obtained, most of which are concentrated in the size range of 1–3 μm, with the highest distribution probability in the size range of 1.5–2.0 μm. The average median particle size is 1.71 μm, accounting for 26.3%. The distribution of particle size presents a standard normal distribution. The particle size distribution of Li_2_TiO_3_ powder is narrow and uniform. The particle size distribution of Li_2_TiO_3_ powder is narrow and uniform. As shown in Figure 2c, three thin and high diffraction peaks of Li_2_TiO_3_ occurred at 2θ of 18.5°, 43.71°, and 63.51°, respectively, and a pure Li_2_TiO_3_ phase with fine crystallites was ultimately obtained after calcining at 600 °C for 1 h. The sample exhibited well-defined peaks assigned to the pure β-Li_2_TiO_3_ phase (PDF card: 33-0831), indicative of the high crystallinity of the sample. No other impurity peaks were found, indicating the high purity of the prepared Li_2_TiO_3_ powder.

### 3.2. Effect of Li_2_TiO_3_ Micro-Powders Content on the Magnetic Field Viscosity of Flowable Tritium Breeder Composite

The Li_2_TiO_3_ micro-powders content in the tritium breeder composite is characterized by Equation (2):(2)$$\varphi_{s}=\frac{V_{s}}{V}$$
where $\varphi_{s}$ is the volume fraction of solid micro-powders, $V_{s}$ is the volume of solid micro-powders, and *V* is the total volume of tritium breeder composite.

Figure 3 depicts the variation of the viscosity of flowable tritium breeder composites in the magnetic fields with different volume fractions of Li_2_TiO_3_ micro-powders. The results show that the flow of the materials displays an apparent MHD effect under the present experimental condition. Firstly, without the magnetic field, the composite containing Li_2_TiO_3_ micro-powders shows larger viscosity than that of the liquid metal. Meanwhile, the viscosity increases with the increasing volume fractions of Li_2_TiO_3_ micro-powders. This phenomenon may be caused by the collision and the friction among the solid Li_2_TiO_3_ micro-powders as well as between the solid Li_2_TiO_3_ micro-powders and liquid metal. Moreover, higher volume fractions of the Li_2_TiO_3_ micro-powders lead to the greater consumption of flow kinetic energy during collision and friction and thereby the larger viscosity. With the external magnetic field, however, the viscosity of composites with various volume fractions of Li_2_TiO_3_ micro-powders increases obviously. Particularly, the liquid metal viscosity has the most remarkably increase, even far exceeding the viscosity of the composites when the magnetic flux density reaches 2305 Gs. This result reveals two opposite influences on the viscosity by adding solid micro-powders into the liquid metal. Firstly, the addition of micro-powders leads to increased viscosity through collision and friction. Secondly, it reduces the MHD effect and therefore decreases the viscosity of the composites in the magnetic field. The above results indicate that the composite containing 10 Vol.% Li_2_TiO_3_ micro-powders, under a magnetic field with 2305 Gs intensity, shows the lowest viscosity.

The effects of magnetic flux density on the viscosity of the flowable tritium breeder composites were further considered, as shown in Figure 4. The results show that the viscosity of the composite with various fractions of solid Li_2_TiO_3_ micro-powders increases monotonously with the magnetic field intensity. As the magnetic flux density increases, the increasing rate of the viscosity of liquid metal becomes obviously faster than that of the composites, whilst the increasing rate of the composites gradually slows down at larger magnetic flux density. When the magnetic flux density is greater than a certain value, the viscosity of the liquid metal is higher than that of the composites, representing that the addition of Li_2_TiO_3_ micro-powders can efficiently reduce the MHD effect and can keep the composites with lower flow resistance under larger magnetic flux density. The changes of the viscosity of the composite with 25 Vol.% Li_2_TiO_3_ micro-powders, in a magnetic flux density reaching about 2000 Gs, gradually tend to be gentle and toward a steady state.

When it is a pure metal, the viscosity and magnetic flux density are in a power of 2 relationship, and the fitting relationship is *η* = 2.31 × 10^−7^B^2^ − 1.98 × 10^−4^B + 0.43 (B is the magnetic flux density; the fitting formula is only used to discuss the numerical relationship of the measured curve. At present, we cannot explain the physical meaning of the formula, so the formula does not represent a strict dimensional relationship. The following Formula and Formula (9) only consider numerical relationship), which is consistent with the results of Zhang Wen et al. [22]. For the relationship between viscosity and magnetic flux density of composites with Li_2_TiO_3_ powder, the formula *η* = A_1_ B^^3^/2 + A_2_ B^^1^/2 + A_3_ is more in line with the relationship between magnetic flux density and viscosity (as shown in Table 1, A_1_, A_2_, and A_3_ are constant). From the different formulas obtained by fitting, it is feasible to use this property to reduce the MHD effect by adding Li_2_TiO_3_ to change the motion of pure liquid metal in the magnetic field.

### 3.3. Effects of Ceramic Micro-Powders on Conductivity of Tritium Breeder Composite

Magnetic fluid pressure drop (Δ*P_MHD_*) is another commonly used parameter to evaluate the fluidity of liquid tritium breeder in magnetic field. Considering the complexity of the real magnetic fluid pressure drop formula, a simplified expression is used according to the theory of Yang et al. [23]:(3)$$\Delta P_{MHD}=k_{p}\sigma_{f}vB^{2},$$
where $k_{p}$ is a constant, and v is the flow rate. The conductivity $\sigma_{f}$ is proportional to the Δ*P_MHD_*, and reducing the conductivity of the tritium breeder composite is obviously the most effective method to reduce the MHD effect [24]. As shown in Figure 5, the conductivity decreases rapidly with the addition of ceramic micro-powders, and the decrease of conductivity follows a certain conductivity formula when ceramic micro-powders are added. Many researchers have studied the conductivity of composites with dispersive second-phase distribution and have derived the calculation formula of conductivity, including the following formulas:

Maxwell conductivity equation [25]: (4)$$\sigma_{M}=\frac{2\sigma_{2}+\sigma_{1}-2\left(1-\varphi \right)\left(\sigma_{2}-\sigma_{1}\right)}{2\sigma_{2}+\sigma_{1}+\left(1-\varphi \right)\left(\sigma_{2}-\sigma_{1}\right)}\sigma_{2},$$

Wiener formulas [26]: (5)$$\sigma_{M}=\sigma_{1}1-\varphi +\sigma_{2}\varphi ,$$

Scarisbrick formulas [27]: (6)σM=σ2φφφ−23C2,
where $\sigma_{M}$ is the conductivity of composites, and $\sigma_{1}$ and $\sigma_{2}$ are the conductivity of insulator phase and conductive phase, respectively. *φ* is the volume fraction of bulk phase, and *C* represents the geometric factor of Formula (6), $\varphi =3C^{2}-2C^{3}$.

Figure 5 presents the calculated and experimental conductivity values according to Formulas (4)–(6). It indicates that the conductivity of the composite decreases with the increase of ceramic micro-powders fractions although the calculated values are greater than the measured ones. Formulas (4) and (5) are applied to the case where dispersed insulating particles are added into the conductive matrix, and Formula (6) is mainly applied to the case where conductive particles are added to the insulating matrix. However, all of above formulas are mainly concerned with the cases where the addition of the second particle is in low volume fraction.

In this paper, however, the insulating particles are added to the conductive matrix in a large amount, so the conductivity values calculated by the above formulas are somewhat much larger than that of experimental ones. By adding Li_2_TiO_3_ micro-powders, the conductivity of tritium breeder composites decreases rapidly from 3.4 × 10^6^ S m^−1^ to 1.62 × 10^5^ S m^−1^, indicating that a much more pronounced effect on the conductivity, compared to that described in Formulas (4)–(6), occurs after adding insulating Li_2_TiO_3_ micro-powder. When measuring the resultant conductivity with different volume fractions, the conductivity of tritium breeder composites achieves the largest value of 9.43 × 10^5^ S m^−1^, while the maximum electrical conductivity of the liquid metal droplets wrapped with polysaccharide microgel is 4.8 × 10^5^ S m^−1^ [28]. Actually, in this investigation, the amounts of insulating particles are large, forming a great deal of interfaces between solid particles and liquid metal. These interfaces are of great importance in carrier scattering. Nevertheless, only the contribution of contact between conductive particles and conductivity of conductive particles is considered in Formulas (4)–(6), while the role of ceramic micro-powder and interface effects are ignored.

### 3.4. Analysis of the Relationship between the Magnetic Fluid Pressure Drop and Kinematic Viscosity of the Flowable Tritium Breeder

Hua et al. [24] calculated the functional relationship between the magnetic fluid pressure drop of the rapidly changing magnetic field and Ha^−1^, fitted the experimental results, and obtained the empirical formula of the magnetic fluid pressure drop:(7)$$\Delta P_{MHD}=Ha^{-1}\sigma vB^{2}l,$$
(8)$$Ha=Ba\sqrt{\sigma v},$$
where σ and v are the conductivity and average velocity of the fluid, B is the intensity of the external magnetic field, l is the length of the pipe wall, Ha is the Hardman constant that can be calculated from Formula (8), and a is $\frac{1}{2}$ of the channel height parallel to the direction of the magnetic field.

The relationship between η and $\Delta P_{MHD}$ of the volume fraction of different ceramic micro-powders is shown in Figure 6. As can be seen from Figure 6, the MHD pressure drop of pure liquid metal is significantly greater than that of the fluid composites. The addition of Li_2_TiO_3_ micro-powder does reduce the MHD pressure drop, and Figure 5 shows that to reduce the conductivity leads to the MHD pressure drop. With the increase of Li_2_TiO_3_ micro-powder, the maximum MHD pressure drop of different volumes of powder decreases gradually. In order to further reveal the relationship between the MHD pressure drop $\Delta P_{MHD}$ and kinematic viscosity η, the curves of the MHD pressure drop $\Delta P_{MHD}$ versus the kinematic viscosity η were fitted. The fitting results are found to conform to the fitting formula:(9)$$\Delta P_{MHD}=m\eta +n,$$
where m and n are fitting constants.

$\Delta P_{MHD}$ shows a linear relationship with kinematic viscosity (when $\varphi_{s}$ is 0, the fitting degree is poor, but when $\varphi_{s}$ is greater than 5 Vol.%, the fitting correlation coefficients is good), and $\Delta P_{MHD}$ increases with the increase of kinematic viscosity, and d$\Delta P_{MHD}$/d*η* decreases with the increase of the proportion of micro-powder fractions (shown in Table 2). This indicates that the $\Delta P_{MHD}$ increasing rate decreases with the increase of viscosity, which indicates that the addition of Li_2_TiO_3_ micro-powders effectively reduces the MHD effect.

### 3.5. Stability Evaluation of Flowable Tritium Breeder

Considering that the prepared tritium breeder is composed of liquid GaInSn alloy and ceramic Li_2_TiO_3_ micro-powders, ceramic particles flowing onto the surface may occur during holding. In this point of view, the variation of the conductivity with time is used as the criterion to judge the stability of the dispersion uniformity. If the conductivity of the fluid remains unchanged with time, obviously, it is indicated that the fluid can be seen as in a stable state. On the contrary, if the conductivity changes with time, then it is considered that ceramic particles flowing onto the surface has occurred. Figure 7 shows the variation of the conductivity of the composite with holding time. When $\varphi_{s}$ is 0, no obvious change of the conductivity can be found, indicating the considerably high stability of the liquid tritium. When $\varphi_{s}$ is 5 Vol.%, the conductivity remains stable within the first 105 min and achieves a stable state after increasing to a certain extent. The above results indicate that the micro-powders can maintain a good mixing state with the liquid metal within 105 min. After that, there are ceramic particles flowing onto the surface, which then continues to maintain a stable state, indicating that a small amount of micro-powders precipitated, but the remaining micro-powders can still maintain a good mixing state with the liquid metal. When volume fraction $\varphi_{s}$ continued to increase to 20 Vol.%, the conductivity remained stable for a certain period of time, and then, after a certain period of time, the conductivity remained unchanged, indicating that there is a certain amount of powder flowing onto the surface. Increasing the $\varphi_{s}$ to 25 Vol.%, it was found that the conductivity increases significantly after maintaining for the first 45 min and then remains stable, indicating that as the volume fraction of added micro-powders is achieved to a certain extent, the mixed liquid becomes unstable. Interestingly, once the mixed materials are stirred by ultrasonic treatment, it can reach the suspended state again and achieve the initial conductivity, indicating a recycling use in practical application.

## 4. Conclusions

In the present study, Li_2_TiO_3_ ceramic micro-powders mixed with liquid GaInSn composite were prepared, in which GaInSn alloy was used to simulate the flow behavior of lithium-based liquid tritium breeder. The effects of volume fraction of ceramic micro-powders on the viscosity and conductivity of the composites in magnetic field were discussed in detail. The main conclusions are as follows:The morphology of Li_2_TiO_3_ micro-powders prepared by sol-gel-hydrothermal method is spherical with high sphericity. The XRD results prove that the obtained Li_2_TiO_3_ micro-powders contain Li_2_TiO_3_phase without impurities;The addition of Li_2_TiO_3_ micro-powders exhibits apparent influence on the viscosity of tritium breeder composite. When Li_2_TiO_3_ powder is added, the fitting formula is *η* = A_1_ × B^^3^/2 + A_2_ × B^^1^/2 + A_3_, and Li_2_TiO_3_ powder changes the motion of pure liquid metal in the magnetic field so that it is feasible to use this characteristic to reduce the MHD effect under the strong magnetic field;The conductivity of the liquid tritium breeder composites decreases rapidly with the addition of Li_2_TiO_3_ micro-powders. By testing the relationship between the conductivity of the composites with different volume fractions powder and time, it was found that the conductivity of the composites increased slightly and then remained stable after static placing for tens of minutes. A small amount of micro-powders precipitate during holding for several of several ten minutes, whereas the remaining micro-powders maintain a uniform distribution with the liquid metal for a long time. However, when the content of micro-powders is large, the conductivity increased with time, indicating that the stability of the mixed fluid is reduced;There is a linear relationship between the magnetic fluid pressure drop $\Delta P_{MHD}$ and kinematic viscosity *η* for different volume fraction of ceramic micro-powders, while the increasing trend of MHD decreased with the increase of viscosity.

In this paper, it is proven that adding Li_2_TiO_3_ powder to liquid tritium breeder could effectively reduce the magnetic fluid effect and form a flowable tritium breeder with low flow resistance under the strong magnetic field, which provides a new idea for the research and development of tritium breeder materials. The study in this paper may point to a way to completely solve many serious problems of tritium breeder materials.

## Figures and Tables

**Figure 1 materials-16-00406-f001:**
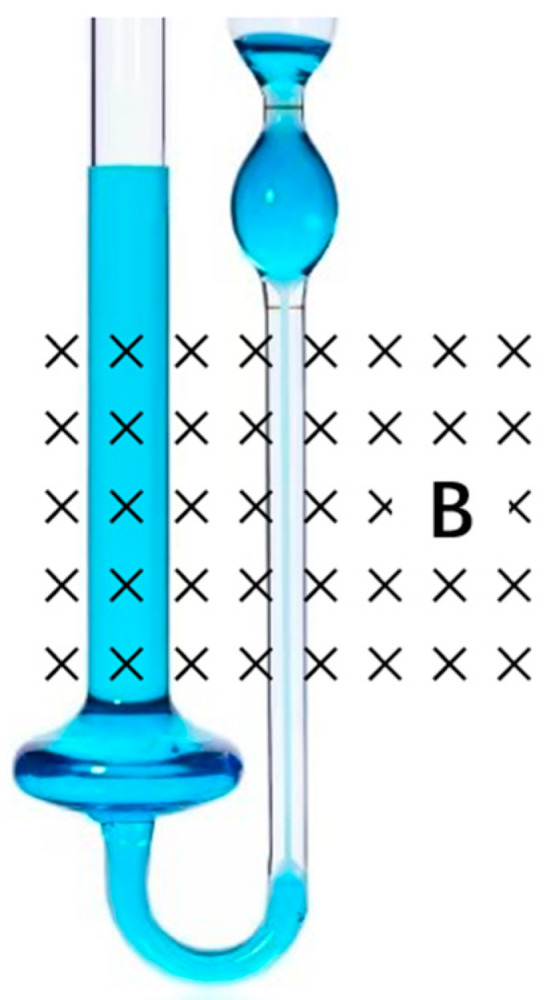
Schematic diagram of the magnetic field viscosity testing device (B is magnetic flux density).

**Figure 2 materials-16-00406-f002:**
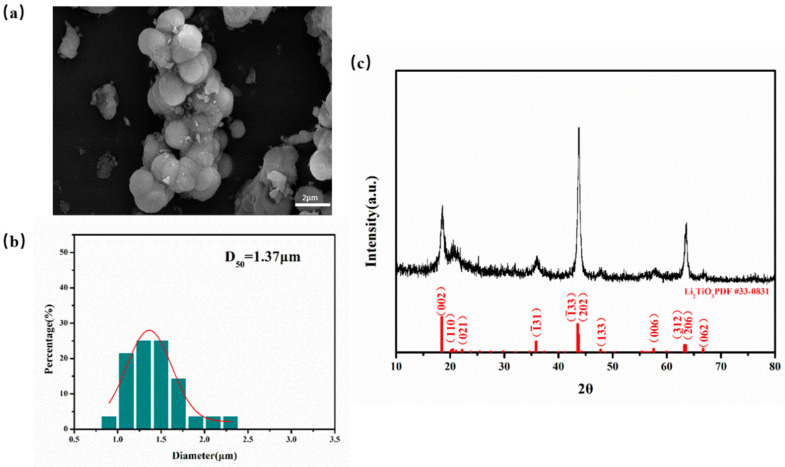
Li_2_TiO_3_ micro-powders prepared by sol-gel-hydrothermal method: (**a**) SEM image; (**b**) particle size distribution histogram; (**c**) XRD pattern.

**Figure 3 materials-16-00406-f003:**
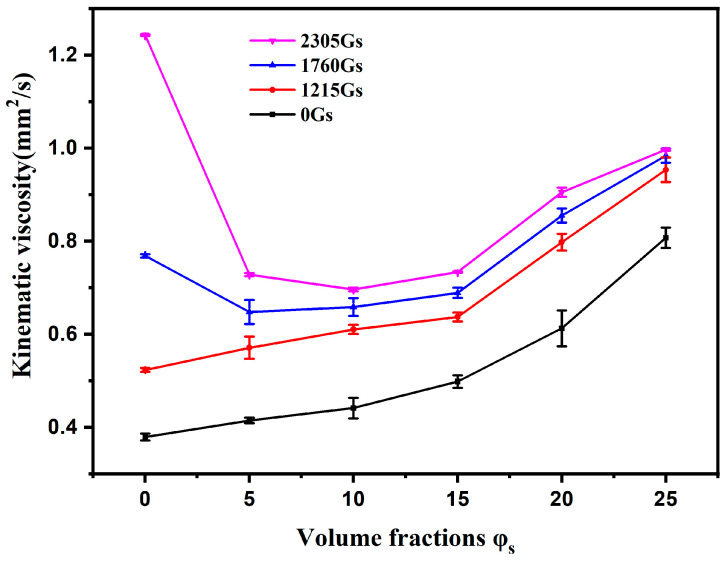
Variation of the kinematic viscosity of tritium breeder composite with different Li_2_TiO_3_ micro-powders fractions.

**Figure 4 materials-16-00406-f004:**
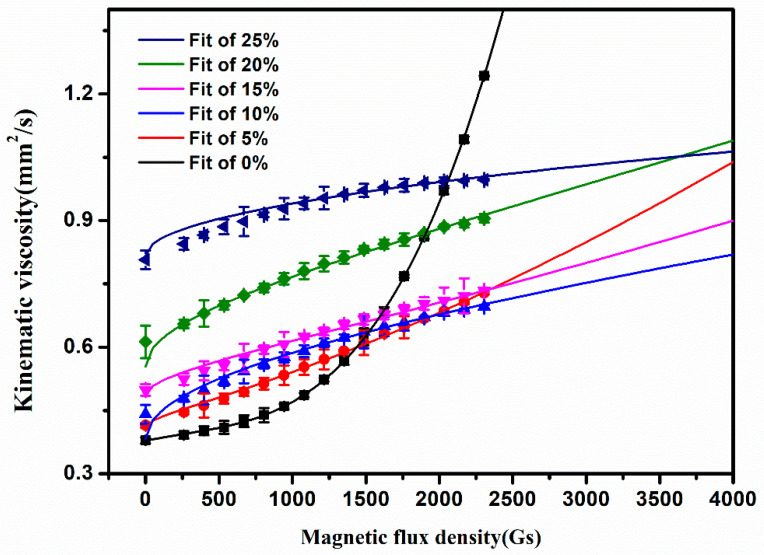
Relationship between magnetic flux density, Li_2_TiO_3_ volume fraction, and kinematic viscosity.

**Figure 5 materials-16-00406-f005:**
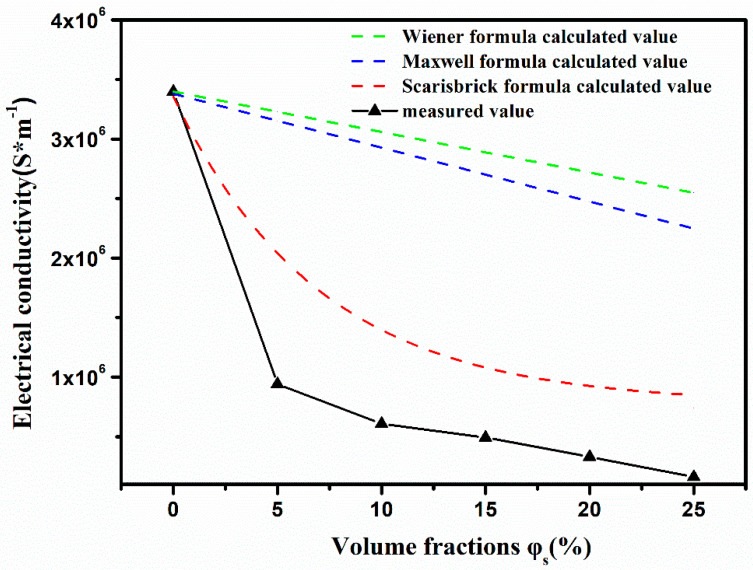
Variation of the conductivity of flowable tritium breeder composite with different volume fraction $\varphi_{s}$.

**Figure 6 materials-16-00406-f006:**
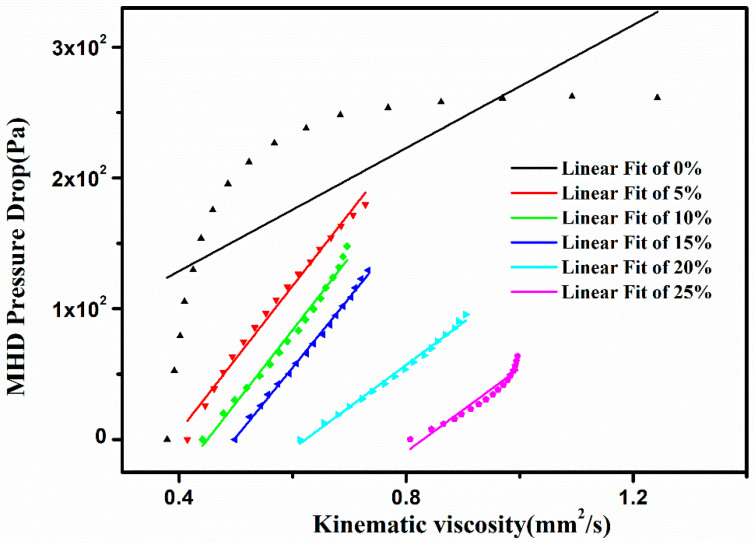
Relationship between magnetic fluid pressure drop $\Delta P_{MHD}$ and kinematic viscosity *η* under different volume fractions.

**Figure 7 materials-16-00406-f007:**
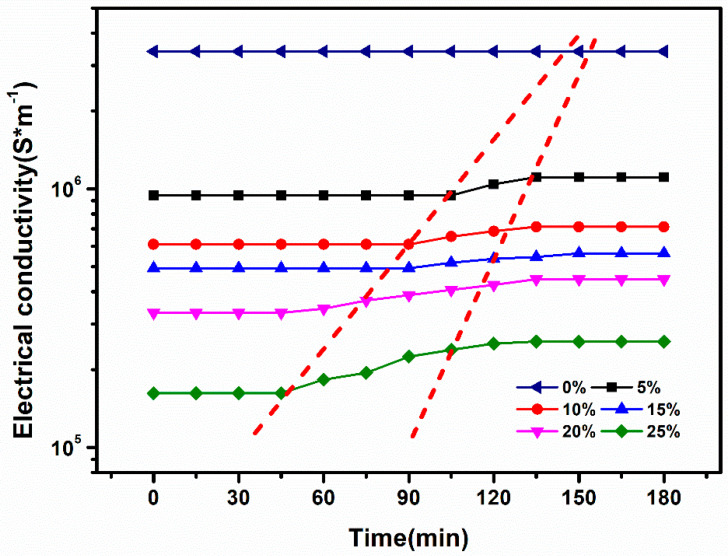
Variation of conductivity with time for different volume fractions of $\varphi_{s}$.

**Table 1 materials-16-00406-t001:** *η* = A_1_ B^^3^/2 + A_2_ B^^1^/2 + A_3_ fitting curve values.

	$\varphi_{l}$	5%	10%	15%	20%	25%
Value						
A_1_		1.93 × 10^−6^	1.58 × 10^−7^	8.09 × 10^−7^	5.82 × 10^−7^	7.64 × 10^−7^
A_2_		2.24 × 10^−3^	6.27 × 10^−3^	3.35 × 10^−3^	6.14 × 10^−3^	7.22 × 10^−3^
A_3_		4.09 × 10^−1^	3.83 × 10^−1^	4.84 × 10^−1^	5.54 × 10^−1^	7.36 × 10^−1^
R^2^		0.9983	0.98973	0.99475	0.99927	0.97323

**Table 2 materials-16-00406-t002:** Parameters related to fitting curve of magnetic fluid pressure drop $\Delta P_{MHD}$ and kinematic viscosity *η*.

	$\varphi_{s}$	0	5%	10%	15%	20%	25%
Value							
m		235.30	557.81	556.65	534.58	321.98	319.62
n		34.64	−216.46	−250.93	−265.67	−200.45	−265.09
R^2^		0.5462	0.98773	0.98748	0.99688	0.99277	0.93173
$d\Delta P_{MHD}/d\eta$		235.30	557.81	556.65	534.58	321.98	319.62

## Data Availability

Not applicable.

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
