# Peer review of "An Innovative Approach to Prepare Liquid-Solid Dual-Phase Flowable Tritium Breeder with Low MHD Effect"

_materials, 2023, doi:10.3390/ma16010406_

Round 1
Reviewer 1 Report
In this study, the authors described the preparation of Li2TiO3 ceramic micro-powders mixed with GaInSn liquid composite, in which GaInSn alloy was used to simulate the flow behavior of lithium based liquid tritium breeder. The effects of volume fraction of ceramic micro-powders on the viscosity and conductivity of the composites in magnetic field were described in detail. It was shown that the morphology of Li2TiO3 micro-powders prepared by sol-gel-hydrothermal method was spherical with high sphericity.The X-ray results prove that the obtained Li2TiO3 micro-powders almost completely contain the Li2TiO3 phase without any other impurities. It has also been shown that the addition of Li2TiO3 micro-powders has an obvious effect on the viscosity of the tritium multiplier composite. At the same time, it was shown that the conductivity of liquid tritium multiplying composites decreases rapidly with the addition of Li2TiO3 micro-powders.
Apparently, the results of this work can contribute to solving some applied problems related to materials for tritium reproduction, and the article itself deserves to be published in the journal Materials.
1. A small remark concerns the form of presentation of the material in the article: each equation must end with either a comma or a dot.
2. In equation 1 it is necessary to give explanations concerning all values.
3. Why is only the value of indicated in the text.
4. It is necessary to explain Fig. 2 c.
5. In lines 113-114, the authors wrote that "Fig. 2(b) shows the particle size distribu-113 tion and the median particle size is 1.71 μm." It is necessary to explain what this means!
6. The authors write (see links 131-132) that "The results show that the flow of the materials displays apparent MHD effect under the present experimental condition." ! Where is it shown and how?
7. How was the effect of magnetic flux density on the viscosity of fluid composites for the production of tritium measured!
8. The authors wrote (see lines 166-167) that “the viscosity and density of the magnetic flux depend on degree 2, and a suitable dependence is = 2.31E-7B2-1.98E-4B+0.43”. But the dimension of is m2/s, but B2 is kg/(s C), not to mention B!
9. The same on lines 169-170.
10. What is the dimension of the coefficients A1, A2,A3 and R?
11. How to explain equation 3? According to the text under equation 3, the value of , is dimensionless! But this is not the case!
12. Formula 6 is correct?
13. Formulas 7 and 8 are correct? What is the dimension of the left and right parts of formula 7.
14. How in formula 9 can the dimension of pressure be equal to the dimension of kinematic viscosity
Only after these points are clarified, it is possible to give a final conclusion about this work!
Reviewer 2 Report
Review Report (Manuscript ID: materials-2117689)
An innovative approach to prepare liquid-solid dual-phase flowable tritium breeder with low MHD effect
In this article, the authors studied the flowable tritium breeder, prepared by mixing the Li2TiO3 micro-powders and liquid GaInSn alloy, where GaInSn alloy is used to simulate the fluid behaviors of lithium-based liquid tritium breeder, forming a type of composite characterized by liquid-solid dual-phase. The present investigation provides a novel insight into the fabrication strategy of tritium breeder materials with low MHD effects.
I suggest that the authors must take into account the following corrections/suggestions:
1. There are some grammatical and typo errors in the paper; the paper should be free from all errors.
2. Polish the abstract by presenting only significant and key outcomes.
3. The author should explain why the study is useful with a clear statement of novelty or originality by providing relevant information in the introduction, such as: Linear and quadratic convection significance on the dynamics of MHD Maxwell fluid subject to stretched surface.
4. Compare your results with the existing literature for different mixtures?
5. The author should add more discussions on graphs and physical features of the obtained results.
6. Recheck Figures 5 and 6 and the results relevant to these two figs.
Concluding Remark:
From my point of view, the results presented are new and interesting. Hence, I recommend the paper for publication. However, the above comments or suggestions must be incorporated before the paper's publication.
Reviewer 3 Report
1. The article is interesting.
2. The literature review is not complete.
3. Research gaps have been clearly identified.
4. The purpose of the research is well defined.
5. In lines 85-88 we read: "Different certain volume fractions of Li2TiO3 micro-powders and liquid GaInSn alloy were mixed, crushed and stirred repeatedly to obtain a uniform liquid-solid dual-phase tritium breeder composite."
The density of the GaInSn alloy is 6.44 g/cm3, the pure Li2TiO3 density is 0.434 g/cm3.
See the work:
https://doi.org/10.1016/j.jnucmat.2009.06.002
http://dx.doi.org/10.1016/j.jnucmat.2013.10.055
With such density differences, we cannot talk about sedimentation. The phenomenon of ceramic particles flowing onto the surface of the alloy occurs here.
6. In line 211 it says conductivity decreases when ceramics are added - this is obvious, but it is important to note.
7. The research results are interesting.
Round 2
Reviewer 2 Report
I accept the paper for publication.